# Protein language models expose viral mimicry and immune escape

**Abstract:**

Viruses elude the immune system through molecular mimicry, adopting their hosts biophysical characteristics. We adapt protein language models (PLMs) to differentiate between human and viral proteins. Understanding where the immune system and our models make mistakes could reveal viral immune escape mechanisms. We applied pretrained PLMs to predict viral from human proteins. achieving a state-of-the-art results (99.7% ROCAUC). We use interpretable models to characterize viral escapers. Altogether, mistakes account for 3.9% of the sequences with viral proteins being disproportionally misclassified. Errors often involve proteins with low immunogenic potential, human specific viruses, and reverse transcriptases. Viral families causing chronic infections and immune evasion are further enriched. Biological and ML models make similar mistakes. Integrating PLMs with explainable AI, we provide novel insights into viral immune escape mechanisms, enhancing strategies for vaccine development and antiviral research.

## 1 Introduction

The interplay between hosts and pathogens involves complex interactions, with viral evasion of the biological immune system being a crucial survival strategy. Viruses often employ molecular mimicry, adopting biophysical characteristics of their host, such as length and sequence composition, to evade immune detection. Understanding these mechanisms is essential for advancing therapeutic interventions, designing effective vaccines, and minimizing adverse immune reactions. Viruses are classified (e.g., Baltimore; 7 groups) based on how the viral mRNA is produced and the nature of their genome (RNA, DNA, and strand properties). The smaller genome size and the smaller number of functional proteins specify the RNA viruses from DNA viruses (Mahmoudabadi and Phillips, 2018). Among the large dsDNA families, Herpesviridae infection is estimated to cover most humans. A unique feature of Herpesviridae is their latency potential. Specifically, after the initial infection, herpes simplex virus (HSV-1, HSV-2), varicella-zoster virus (VZV), human cytomegalovirus (HCMV), Epstein-Barr virus (EBV) and others may enter a dormant phase. These herpesviruses developed mechanisms to evade the host's immune system (Cohen, 2020). Human genomes include endogenous retroviruses (ERVs) which are remnants of ancient infectious retroviruses. While they are mostly dormant, they can be activated upon various stimuli and may limit the spread of current viral pathogens (Srinivasachar Badarinarayan and Sauter, 2021).

Viral and human proteins are expected to express unique features that reflect their distinct evolutionary origins. Many of the viruses impacting human health, such as human immunodeficiency virus (HIV), and influenza A (sessional flu), carry only a handful of proteins (Brandes and Linial, 2016). The minimal functional set of viral proteins is essential to complete their life cycle. All viruses include proteins that support host cells' entry, replicating enzymes, assembly components, and structural components (e.g., envelope, capsid subunits). These viral-specific features are mostly absent among human proteins (Mahmoudabadi and Phillips, 2018), but cases in which human genetic material is detected in viruses and vice versa were reported (Rappoport and Linial, 2012). Importantly, viruses may have proteins specifically for evading immune responses. A classic example of viral protein mimicry involves HIV and its mimicry of host cell receptors. HIV primarily infects CD4-positive T lymphocytes and macrophages through its envelope glycoprotein, gp120. Gp120 structurally mimics the host cell receptor CD4. The dynamic interaction between gp120 and CD4 allows HIV to evade immune surveillance by masking key epitopes and thus prevent effective neutralizing response against the virus. Epstein-Barr virus (EBV) mimicry underlying cases of cancer such as Burkitt and Hodgkin lymphoma. For example, Epstein–Barr nuclear antigen 1 (EBNA1) contains regions that mimic host proteins and consequently manipulate host gene expression that is involved in the immune response. Structural similarity between EBNA1 and host proteins may lead to cross-reactivity of antibodies which is the hallmark of several autoimmune diseases (Poole, et al., 2006). Additional mechanisms for exploiting cellular machinery include priming of pathogenic responses by auto-antigens are known (Thomas and Olsson, 2023.). These mechanisms are evolutionarily optimized through sequence adaption (Bahir, et al., 2009).

Pretrained, deep-learning (DL) language models have boomed in recent years (Ofer, et al., 2021). Pretraining is a process wherein a machine learning model learns to understand and predict sequence patterns without supervision, before fine-tuning on specific tasks, while enjoying strong learned priors and improved performance on the downstream tasks. Biology is no exception, with biological sequence language models showing promising results with DNA, RNA, and protein sequences. Protein Language Models (PLMs), including ProteinBERT, ESM2, CARP, and Prot-T5 have shown state-of-the-art performance in predicting structure, function, mutation, variant effects, unsupervised anomaly detection and generating novel proteins, from the sequence (Brandes, et al., 2023; Brandes, et al., 2022; Elnaggar, et al., 2022; Nijkamp, et al., 2023; Rives, et al., 2021; Yang, et al., 2024). However, analogies, or dissimilarities, between these models and comparable biological mechanisms involved in biological sequences have not been well studied. In the case of natural languages and computer vision, it has been observed that learned model representations can alternatively mimic biological

mechanisms (e.g., in learning low-level visual features and filters), or can have very different representations, such as focusing on textures rather than shapes (Geirhos, et al., 2018; Zeman, et al., 2020). In vision, challenging or "adversarial" examples may fool computer vision models while being obvious to the human eye, or may be challenging to both models (Elsayed, et al., 2018).

In this study, we investigate the performance of PLMs in distinguishing between viral proteins and their hosts while focusing on human-virus protein sequences. Specifically, we demonstrate that PLMs and the immune system both have difficulty accurately identifying viruses that are adept at mimicking host proteins. Our approach offers novel insights into the mechanisms underlying viral escape. Our findings reveal a striking parallel between mistakes made by PLMs in classifying proteins and those encountered by the failures of the natural immune system of the host. Mitigating the impact of viral infections on human health can benefit from inspecting the model's success and faulty classification in cases of viral mimicry.

## 2    Methods

### 2.1. Protein datasets
All reviewed human proteins and virus proteins with a known, vertebrate host were downloaded from SwissProt within the UniProtKB database (https://www.uniprot.org) (Status: Reviewed, non-fragment, Nov 2021), along with their Uniref50 and 90 sequence similarity clusters, and annotations: UniProt keywords, name, taxonomy, virus-host, and length.

Duplicate sequences at the UniRef90 level were dropped to reduce redundancy. Proteins longer than 1,600 were excluded. The dataset was shuffled and partitioned by UniRef50 clusters into a training subset (80%) and a test subset (20%). Sequences sharing the same cluster (i.e., greater than 50% sequence identity) were always disjoint between train and test sets. Protein-level embeddings were downloaded from UniProt. Virus family, genus, and Baltimore classification were downloaded from ViralZone (Masson, et al., 2012).

### 2.2. Pretrained deep language models (ESM, T5)
ESM2 is a deep learning architecture, based on the BERT Transformer model (Lin, et al., 2023). It was pretrained on the UniRef50 dataset to predict masked-out amino acids (tokens). It can efficiently represent amino acid sequences and has shown good performance across different protein predictive tasks. We used different-sized ESM2 models. We use mean pooling for extracting a sequence-level representation of each protein. This approach has been shown to yield a good representation in sequence-level problems. Specifically, the final dense layer of the chosen model is taken and its representation of each token (an individual amino acid) in the sequence is averaged over all tokens in the sequence. This representation can be followed by training on a specific task. We also downloaded pre-extracted protein level embeddings, from UniProt (https://www.uniprot.org/help/embeddings) ("T5"). These embeddings were derived from "prottrans_T5_xl_u50", a T5 Transformer architecture PLM model, with 3 billion parameters (Elnaggar, et al., 2022). These are used as input features for training downstream, non-deep ML models.

### 2.3. Human-virus model training and implementation
Models were trained to predict if a protein is from a human or a virus. Performance was evaluated on the test set. The pretrained DL ESM2 models were fine-tuned in PyTorch, using the HuggingFace transformers library (Wolf, et al., 2019). DL models were trained on a 16GB, 4080 NVIDIA GPU with 8-bit Adam optimizer, fixed learning rate 5e-4, 16-bit mixed precision training, LoRA, batch size 16, max sequence length 1024, and cross-entropy loss for 3 epochs. For finetuning the DL models, we used LoRA (low-rank adapters), a parameter-efficient fine-tuning method for transformers, implemented using peft (Hu, et al., 2021). LoRA adapters were attached to all linear and attention layers, with LoRA $r$=8, $\alpha$=8, dropout $p$=0.
The ESM models were trained using only sequences. Scikit-learn implementations and default hyperparameters were used for logistic regression (LR) and histogram gradient boosting decision trees (GBT) models (Pedregosa, et al., 2011). The length baseline is a LR model trained only on sequence length. The amino acid (AA) n-grams model is a GBT trained on length and AA n-gram frequencies (Bahir, et al., 2009) features, as a strong baseline (Michael-Pitschaze, et al., 2024; Ofer and Linial, 2015).

### 2.4. Finding and analyzing model mistakes
Following model training and evaluation, in order to analyze model errors and misclassifications, we performed an additional stage of extracting predictions over the whole dataset. We used the simple, static embeddings model ("Linear-T5"). We rejoined the train and test data together and extracted predictions for the combined data, using 4-fold, group-stratified cross-validation, with retraining in each split. Sequences were again partitioned by UniRef50 clusters. Finally, for each test split predictions that differed from the ground truth were marked as "mistakes". Namely, if a model predicted a human protein for being a virus (abbreviated H4V), or vice-versa (V4H).
To understand the mistakes made by the human virus model, we ran separate error analysis models, where the target was defined as whether our original model had made a mistake. Multiple partitions of the data were analyzed separately: the entire dataset (25,117 sequences), a human-only subset (18,418 sequences), and only viruses from genera with a human host (3,915 sequences). New features were extracted using the SparkBeyond autoML framework (Ofer and Linial, 2022). Inputs included the protein sequence, length, taxonomy, name, UniProt keywords, virus-host species, and Baltimore classification, but not embeddings. Features were automatically ranked by their support (number of examples), lift (likelihood of a target class under the distribution induced by the feature, using an optimal binary split), and mutual information with the target, and selected for explanatory value.

## 2.5. Dimensionality reduction of features

t-SNE (t-distributed Stochastic Neighbor Embedding) dimensionality reduction was used to visualize the embeddings and map it to a low-dimensional space (McInnes, et al., 2018; Van der Maaten and Hinton, 2008).

## 2.6. Model performance

To assess the different model's classification performance, we use the common metrics of AUC (area under the receiver-operating characteristic curve), precision, recall, accuracy, and log-loss. AUC reflects the model's discriminatory ability across thresholds and the trade-off between sensitivity and specificity, with an AUC of 50% reflecting a random predictor and 100% a perfect one.

## 2.7. Immunogenicity datasets and scores

To test immunogenicity the IEDB, Class-I Immunogenicity predictor was used (http://tools.iedb.org/immunogenicity) for estimating immunogenicity scores (Vita, et al., 2019). Due to interface limitations, we ran on 200 samples for each combination of human/virus and the model mistake (True/False), for a total of 800 samples.

## 3    Results

Our objective was to evaluate the performance of protein language models in distinguishing between human and viral proteins and to analyze the types of errors these models make. This aims to deepen our understanding of how computational models can mirror biological processes, particularly in the contexts of taxonomic classification and immune evasion.

## 3.1. Human virus models

**Table 1** shows models performance on the held-out test-set. While the Amino Acid n-grams model, using only sequence length and amino acid combinations achieves good separation (91.9% AUC), the PLM-based models reach 99.7% AUC and ~97% accuracy. We observe that the larger the underlying DL model, the better the results. Training a logistic regression ("linear") or tree model atop the static, T5 embeddings ("T5") is competitive with fine-tuning the ESM2 model, while requiring negligible computation, and is more stable to train and reproduce.

**Table 1.** Human-virus classification models performance

| Model | AUC | Accuracy | Precision | Recall | Log-Loss |
|---|---|---|---|---|---|
| Length baseline | 61.97 | 78.50 | 78.50 | 78.50 | 0.52 |
| AA n-grams | 91.95 | 88.49 | 88.49 | 88.49 | 0.28 |
| ESM2 8M | 98.09 | 94.72 | 92.15 | 92.33 | 0.20 |
| ESM2 35M | 98.69 | 95.83 | 93.81 | 93.92 | 0.18 |
| ESM2 150M | 99.26 | 96.99 | 95.54 | 95.48 | 0.12 |
| ESM2 650M | **99.67** | **97.86** | **96.85** | 96.68 | 0.09 |
| Linear-T5 | 99.56 | 97.57 | 97.57 | 97.57 | **0.06** |
| Tree-T5 | 99.65 | 97.7 | 97.7 | **97.70** | **0.06** |

AA, Amino Acids; T5, T5 model embeddings from UniProt. Values are in %. Bold: best performance.

## 3.2. Error analysis models insights

In our analysis of errors made by the most stable human-virus model (**Table 1**, Linear-T5) there is an overall mistake rate of 3.9% over the joint dataset. Overall, models more frequently misclassify viruses as human proteins (abbreviated V4H) than the other way around (H4V). Altogether, 9.48% of viral proteins are misclassified (635/6,699), as opposed to only 1.87% of human proteins (H4V, 345/18,418), a five-fold difference.

**Table 2.** Overall features of mistaken proteins

| Features | Mistake rate (%) | # proteins | Lift |
|---|---|---|---|
| "Adaptive immune" keyword | 60.5 | 46 | 15.5 |
| Endogenous retrovirus | 30 | 40 | 7.7 |
| Oncogene keyword | 19.3 | 393 | 4.9 |
| Sequence length < 170 | 12.1 | 4539 | 3.1 |
| Virus | 9.4 | 6699 | 2.4 |
| Name "putative" | 8.7 | 1050 | 2.2 |
| Few keywords (< 8) | 8.8 | 3326 | 2.2 |

Lift is frequency of mistakes relative to the overall background

**Table 2** lists features associated with an elevated fraction of mistakes. We calculated the lift to determine the enrichment relative to a prior mistake background. The presence of endogenous retroviruses in the human genome results in a high rate of H4V mistakes, and are arguably not "human". Endogenous retroviruses are of viral origin and have become embedded in the human genome. Similarly, being short or poor in annotations increases the number of mistakes **(Table 2).** Viral proteins annotated as involving the adaptive immune system are also extremely elusive, reflecting their evolved roles.

## 3.3. Virus errors analysis

We evaluated the model's tendency to err based on host specificity. We found mistakes are higher for genera of human-specific when compared to genera of viruses that infect vertebrates. Specifically in viruses, the 3,915 proteins from genera with a human host (i.e., evolutionarily related to human targeting viruses) are mistaken more (10.2%) than the overall rate for (any vertebrate-host) viruses, while the 2,787 Human-host viruses are even more "confounding" (11.6%), as might be expected thanks to evolutionary adaptation.

**Table 3** shows biases between the main classification groups (Baltimore groups I-VII), based on ViralZone. We observed large (up to ~30-fold between dsRNA and dsDNA-RT) differences in mistake rates. We do not attribute this to the nature of the genetic materials (RNA or DNA). Instead, reverse transcriptase (RT) replication-dependent viruses consistently show more V4H mistakes.

**Table 3.** Mistakes by Baltimore class

| Baltimore Group | Genetic material | # families | Mistake. rate (%) | # proteins | Lift |
|---|---|---|---|---|---|
| VII | dsDNA-RT | 1 | 34.2 | 108 | 3.6 |
| VI | ssRNA-RT | 1 | 19.5 | 666 | 2.0 |
| II | ssDNA | 3 | 13.1 | 129 | 1.3 |
| I | dsDNA | 13 | 8.2 | 4421 | 0.8 |
| IV, V | ssRNA | 28 | 8.1 | 1017 | 0.8 |
| III | dsRNA | 4 | 0.8 | 358 | 0.1 |

Based on ViralZone Baltimore group statistics. Lift is relative to prior mistake rate of viruses (9.47%)

### 3.4. Latent structure embeddings clustering

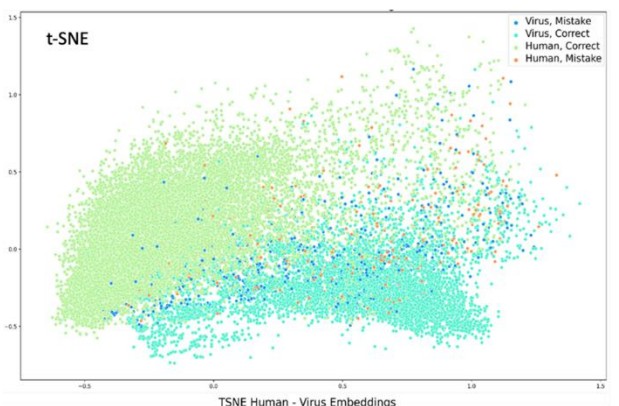

**Fig. 1. Human-virus t-SNE embeddings.** 25,117 sequences shown, including 18,418 from the human proteome.

T-SNE was used to visualize the sequences' embeddings. (**Fig. 1**). While human and viral proteins cluster somewhat separately, the mistakes V4H and H4V are distributed, and no sign of aggregation is observed. While the human-virus protein features and embeddings are different both in terms of sequence composition, length, and sequence embedding, they are not linearly separable, and mistakes are widely distributed throughout the latent space, as opposed to forming distinct clusters (e.g., of endogenous retroviruses).

There are large differences in mistake rates between major viral genera and families, as illustrated in Table 4. Many viruses that cause long-term or life-long diseases are prominent. For example, hepatitis E (liver disease), HIV (AIDS), HPV (Papilloma) and more. A full list of families and genera is provided in our repository. **Supplementary Table S1** provides a list of all human and viral sequences (total 25,117), model prediction scores and mistakes (980).

### 3.5. Immunogenicity analysis

Immunogenicity scores reveal differences in how the immune system, mirrored by PLMs, is sensitive to host versus viral proteins. In **Fig. 2**, we analyze the immune epitope database (IEDB) predicted immunogenicity score distributions across four distinct combinations: virus (**Fig. 2A**) or human proteins (**Fig. 2B**), with or without mistakes (**Fig. 2**, top and bottom, respectively).

**Table 4**. Mistakes by Virus Family (V4H).

| Viral Family | Baltimore Group. | Disease | Mistake Rate (%) | Support | Lift |
|---|---|---|---|---|---|
| Hepeviridae | IV | Hepatitis | 44.4 | 9 | 4.7 |
| Hepadnaviridae | VII | Hepatitis | 34.3 | 108 | 3.6 |
| Circoviridae | II | CNS infection | 33.3 | 27 | 3.5 |
| Polyomaviridae | I | Cancer | 30.7 | 62 | 3.2 |
| Picornaviridae | IV | Nose/Throat | 28.6 | 7 | 3.0 |
| Retroviridae | VI | Cancer/AIDS | 19.5 | 666 | 2.1 |
| Polydnaviriformidae | I | N.A. | 18.4 | 49 | 1.9 |
| Arteriviridae | IV | N.A. | 18.2 | 22 | 1.9 |
| Papillomaviridae | I | Cancer | 14.2 | 520 | 1.5 |
| Caliciviridae | IV | Intestines | 13.8 | 29 | 1.5 |
| Paramyxoviridae | V | Mumps | 12.9 | 124 | 1.4 |
| Anelloviridae | II | Immune Supp. | 11.5 | 52 | 1.2 |

Representative disease by virus unique to human as their host. Lift is relative to the prior mistake rate of viruses (9.47%). N.A. No human specific genus.

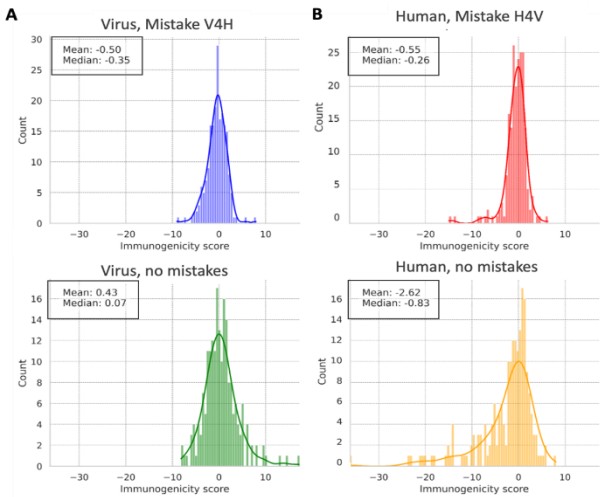

**Fig. 2. Distribution of the IEDB (immune epitope database) immunogenicity scores. (A)** Viral proteins with and without mistakes (top and bottom, respectively). **(B)** Human proteins with and without mistakes (top and bottom, respectively). Note that the mean and median scores for the proteins that were mistaken by the DL model are different.

The IEDB immunogenicity scores reflect the propensity of a sequence to elicit an immune response, with higher scores indicating greater potential for detection by T-cells. The observed distributions suggest differential immune recognition for viral versus human proteins (Vita, et al., 2019), which can be essential for understanding viral escape mechanisms.

Viral proteins typically exhibit more extreme immunogenicity scores compared to human proteins. This aligns with

the notion that viral proteins, especially those mimicking host sequences, have in competition with the immune system co-evolved to present extremes of immunogenicity, either being highly recognizable threats or effectively camouflaged to immune surveillance.

Notably, the subset of mistakes where the PLM mistakenly identified viruses as human (virus, mistakes, V4H; **Fig. 2A,** top) demonstrate a tight clustering of scores around lower values. This indicates a subset of viral proteins that effectively mimic host immunogenicity patterns, blurring the lines for both biological and algorithmic detection.

Conversely, the true-positive viral detections (virus, no mistakes, **Fig. 2A,** bottom) generally show a broader and higher range of scores, supporting the immune system's ability to recognize and respond to these viral entities more robustly. For human proteins, mistakes by the PLM (**Fig. 2B,** top) display a score distribution pattern that suggests a false flag of immunogenicity, possibly reflecting sequences with viral-like properties that could be remnants of ancient viral infections or endogenous retroelements.

An intriguing observation is that irrespective of origin, the proteins falsely identified—whether viral or human (H4V, V4H) share more commonalities in their immunogenicity profiles with each other (with mean immunogenicity scores hovering around -0.5 to -0.55) than with their correctly categorized counterparts. This pattern unveils a potential blind spot in both biological and algorithmic recognition systems, suggesting that certain protein features associated with immune evasion are consistently challenging to discern. This insight not only sheds light on the intricacies of host-pathogen interactions but also marks an area for improving the accuracy of PLMs in biomedical applications such as vaccine development and antiviral drug design.

## 4   Discussion

The main finding is the capability of PLMs to distinguish between human and virus proteins. While high, it is far from perfect. Inspecting the nature of the cases in which the computational model failed highlights instances where the human immune system also fails to recognize and eliminate those same virus pathogens. The failure to eradicate latent viruses has been proposed as a potential cause of autoimmune diseases (AIDs). For example, multiple sclerosis (MS) and rheumatoid arthritis (RA) are linked to viruses through molecular mimicry of Epstein-Barr virus (EBV) and herpesvirus infections. Focusing on the remote relatedness of viral and human proteins revealed sequences and structural similarities that were attributed to virus-induced AIDs (Begum, et al., 2022).

Our work is the first (to our knowledge) to use supervised PLMs and interpretable ML for this task, as well as establishing state-of-the-art results. Previous work in distinguishing between human and virus proteins used an anomaly detection approach (Michael-Pitschaze, et al., 2024). It did not extract explanations for mistakes, nor the effects of taxonomy (e.g., evolutionary adaptation to humans as hosts). However, human endogenous retroviruses were highlighted as anomalies when comparing human/virus

sequences and were associated with H4V mistakes in this study. Language models for viral escape were presented where mutations that occurred altered the meaning while maintaining grammaticality. The analogy was applied to influenza, HIV, and SARS-CoV-2 proteins (Hie, et al., 2021). Lastly, DL embeddings and attention can be challenging to interpret, especially in biology (rather than reading text or viewing images). The interpretability approach used here is also novel in its use of an external autoML model to extract many different features to explain the mistakes made by the investigated DL model. This approach employs not just a different feature set, but also features which could not be used by any parent model, such as class labels, and label-specific partition (e.g., viral Baltimore class) which would cause target leakage if used in the primary model. The concept underlying the success in classifying viral and human represented proteins, was also applied in the task of identifying genes as being targets for miRNA regulation with a great success (Ofer and Linial, 2022).

We observed that overall, algorithmic models are surprisingly aligned to biological ones, in terms of the types of "adversarial" sequences that both find challenging.

These results emphasize the complexity of immune recognition and highlight the immunogenicity landscape's role in the ongoing molecular arms race between host and viral proteins. Moreover, they provide evidence that PLMs and taxon classification pretext tasks might serve as a proxy for studying and predicting immune evasion, potentially aiding in the prediction and design of vaccine candidates (Hie, et al., 2021; Ofer, et al., 2021).

The characterization of specific viral proteins that were misclassified could help future work on the evolutionary strategies that these specific viruses developed to evade the host immune system, such as suppressing the adaptive immune system, or strategies that may have oncological impact and relevance.

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

## Appendices

**Model performance metrics definition:**

Precision, recall and accuracy metrics are calculated as follows:

$$\text{Precision} = \frac{TP}{TP + FP}$$

$$\text{Accuracy} = \frac{TP + TN}{TP + TN + FP + FN}$$

$$\text{Recall} = \frac{TP}{TP + FN}$$

where TP: true positive, FP: false positive, FN: false negative, TN: true negative.
AUC and Log-loss were calculated using scikit-learn. Precision and recall used macro averaging (arithmetic mean across all classes).

**LoRA method overview**
During fine-tuning, LoRA replaces updates ($\Delta\mathbb{W}$) to the original weight matrix ($\mathbb{W}$) with the decomposition of $\Delta\mathbb{W}$ into two low-rank matrices *A* and *B*. Only the weights of *A*, *B* are updated during back-propagation. After training has been completed, *A* and *B* can be multiplied to yield $\Delta\mathbb{W}$ and to update the underlying weight matrix. This approach uses ~1% as many trainable parameters as regular fine-tuning, allowing for faster training times, larger models, and batch sizes, and has outperformed regular fine-tuning in some cases (Hu, et al., 2021).

**Human-virus embedding clustering**

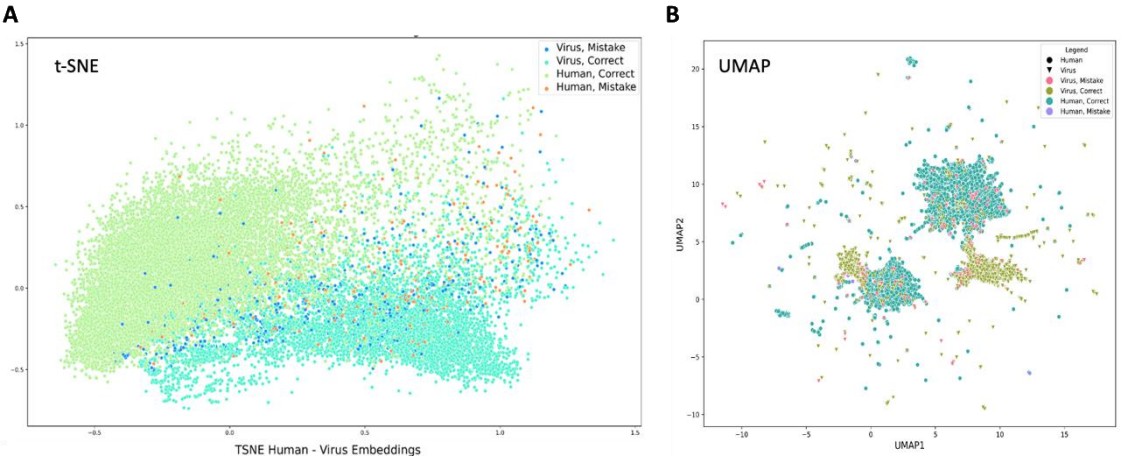

**Fig. 1. Human-virus embeddings**. **(A)** t-SNE embeddings. **(B)** Supervised UMAP (Uniform Manifold Approximation and Projection)  embeddings, based on human/virus labels. Supervised UMAP uses the target label as part of the dimensionality reduction. 25,117 sequences were shown, including 18,418 from the human proteome.

**Biological insights on mimicry**
We illustrate cases that best explain the underlying immune escape mechanisms.

**Fig. 3A** shows a taxonomy view for interleukin-10 (IL-10), an immunosuppressive cytokine produced by various immune cells, including T cells, B cells, macrophages, and dendritic cells. The sequence is part of the UniRef50 P0C6Z6 cluster that is represented by IL-10-like protein in the Epstein-Barr virus (EBV, also called human herpesvirus 4, HHV-4). The model predicted this protein as human. Indeed, viruses and humans utilize IL-10 to modulate immune responses for their benefit. IL-10 It is used as a key player in inhibiting pro-inflammatory cytokine induction in viruses such as Epstein–Barr

virus (EBV), equine herpesvirus (EHV), and cytomegalovirus (CMV). The family of IL10 is abundant in Biletaria (2k proteins, green) but viruses occupy about 10% with similar sequence, structure and function.

The IL-10 is an example of a viral protein that acquired a sequence whose protein product can attenuate the host immune response. Interestingly, while not all 233 listed viral proteins infect humans, the conservation of sequence and structure is extremely high. For example, the Parapoxvirus genus (Fig. 3B) includes Orf viruses (ORFV, 97 proteins) that can lead to human disease through contact with an infected host (e.g., cattle) and a shutdown of the human immune response via the viral IL-10 homolog.

The UniProt50 representative P11364 called Viral T-cell receptor beta chain-like (Feline leukemia virus; 321 aa), matched two Immunoglobulin-like folds (Ig) domains according to InterProScan: InterPro Immunoglobulin (Ig) V-set (IPR013106) and C1-set (IPR003597). Both domains were identified in viruses, bacteria, and eukaryotes (Fig. 3C). The V-set resembles the antibody variable domain as appears in T-cell receptors (e.g., CD2, CD4, CD80, CD86) but also the major protein PD1. Fig. 3D shows the phylogenic partition of metazoan and viruses and their diversity and number of species. Applying structural modeling (SwissModel and AlphaFold2) for the viral sequence identified an exceptional 3D similarity with human and other mammals' T-cell receptors (TCR). In general, these domains are found in either cell surface or soluble proteins. In all cases, the role of the domain is in recognition, binding, or adhesion processes. The second domain represents C1-set domains, which resemble the antibody constant domain. C1-set domains are found almost exclusively in molecules involved in the immune system, such as in light and heavy chains of Ig, in the major histocompatibility complex (MHC) class I and II complex, and various T-cell receptors. We conclude that the role of Ig-like domains in cell-cell recognition, cell-surface receptors, muscle structure, and importantly to form the T cell receptor of the immune system makes this sequence critical for competing with the host immune system. Importantly, several of the reported shared function of viral and human proteins were only identified through structural relatedness (i.e., using HHPred, SwissModel or AlphaFold2) and were not evident through a sequence matching protocols (e.g., BLAST search).

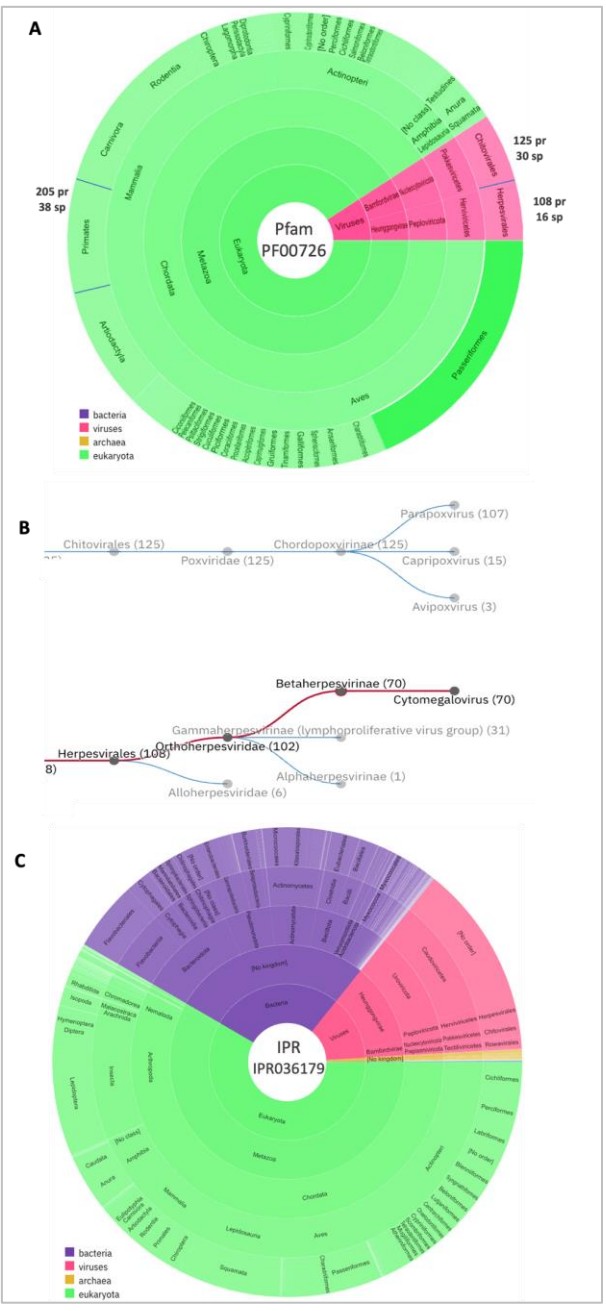

**Fig. 3. Pfam taxonomy view for IL10 (PF00726). (A)** A dominant occurrence in Biletaria (2k proteins, green) and viruses partitioned as 90% and 10%, respectively. There are 205 IL-10 protein family in primates, and 233 such viral proteins (red). **(B)** IL10 viral proteins. The viruses are split to the order Herpesvirales (108 proteins) including 70 representatives of the genus cytomegalovirus (CMV) that infects humans. The other major viral group belongs to the family Poxviridae (125 proteins) including smallpox and various viruses causing zoonotic diseases. **(C)** Example ssRNA-RT of the order Retroviridae (321 amino acids) that mistakenly identified as a human protein. The sequence matches two copies of the Immunoglobulin (Ig) fold. The view shows the number of species of Ig superfamily in eukaryotes, bacteria, and viruses. Such Ig InterPro family (IPR036179) includes 235k of metazoan proteins. The taxonomy pie (A, C) shows a resolution of a 5-ring view color-coded by the major kingdoms.

