# OpenReview forum: "Protein language models expose viral mimicry and immune escape"
_ICML.cc/2024/Workshop/ML4LMS — ML4LMS Poster_

### Official Review · Reviewer_PNqJ · 2024-06-02
**Review of "Protein language models expose viral mimicry and immune escape"**

**Rating:** 4
**Confidence:** 3

**Review:**

The authors fine-tune PLMs for the task of predicting if a protein sequence comes from a virus or a human.

Strengths:
* The manuscript is generally well-written and contains a detailed description of the model architectures and datasets used
* They show that they are highly performant on their own benchmark
* They include appropriate baseline models (Length Baseline, AA n-gram) in evaluating it.
* They use simple statistical associations via the autoML framework to note which features are associated with classifier errors, offering meaningful insight into which sequences their model struggles with.

Weaknesses:
* While applications to this task of distinguishing viral proteins is novel, PLMs have already been shown to be highly performant for diverse tasks from predicting protein structure to function. At its face, it is not surprising that PLMs could do well at this task, and the authors do not provide any motivation suggesting that PLMs might struggle on this task. Therefore, this study feels limited in its novelty and impact.
* The benchmark appears to be too easy as most of their models achieve ~99% AUC. While I don't see any glaring flaws in its construction, 50% sequence similarity cutoff is relatively weak. A 30% cutoff would've provided a more challenging evaluation and a better idea of generalization.
* There are no error bars, and many of their models perform very similarly to each other. This makes comparisons about which is "best" somewhat useless.
* While the authors emphasize that it is helpful to see that the model makes errors on the same sequences that tend to have better immune escape, this insight is not particularly surprising. The viruses evolve to look closer to human proteins (while preserving their viral function), so of course their sequences are more similar than humans.
* The explainability is not as a function of sequence, but rather features that were already available in the online databases. While this addition adds to the quality of the paper, its explainability is still limited to the quality of the features and cannot find new features. Moreover, the autoML framework basically is performing statistical associations (to my understanding after googling because it isn't explained in the paper). This is helpful and I enjoyed thinking about the results; however, this explainability method, finding statistics between groups using a lightweight ML method, is therefore not as novel as they claim in their discussion.
* The authors appear to use the terms "interpretability" and "explainability" interchangeably. Please be careful not to mix them as they have very distinct meanings. In this case, the methods are explainable as they assign post-hoc explanations of the model's failures rather than any mechanistic information about what made the sequence harder to interpret.

Opportunities to Improve:
* These models are fairly lightweight and the dataset is relatively small. Training multiple versions and getting error bars would improve analysis of which model is better.
* Increasing the difficulty of your benchmark by more strict sequence similarity cutoffs would again allow for more meaningful comparisons.
* Look for opportunities to apply the insights/model to important tasks in the field, especially in predicting immune evasion / designing vaccine candidates as the authors suggest.

Overall Review:
This is a good example of how PLMs are powerful new tools for analyzing protein structures, and the authors do a thorough job of explaining the model's failures through explainability techniques. Nevertheless, I don't feel that this work represents a meaningful advancement to the field that merits being showcased at the conference. I encourage the authors to look for ways to expand the impact of their findings, especially by connecting their findings to predicting immune escape as they allude to.

---

### Official Review · Reviewer_WrXd · 2024-06-08

**Rating:** 8
**Confidence:** 3

**Review:**

## Summary

In this manuscript the authors leverage embeddings from protein language models (PLMs) to discriminate between human and viral proteins.
They propose to either fine tune such PLMs or to train a simple additional model on top of the embeddings (logistic regression or random tree).
They empirically show the ability for both of these approaches to reach high accuracy.
They then investigate the 'viral escapers' (i.e. false negatives) and conclude that these are similar to the ones from the human immune system.
In particular, proteins with low immunogenic potential and human specific viruses tend to be wrongly classified as human proteins by the models.

## Strengths & Weaknesses

I enjoyed reading this paper, although it was at times referring to concepts in biology that I was not too familiar with. The targeted audience has likely more of a biology background than an ML one.
Overall, I found the discussion particularly interesting, highlighting the complexity of immune recognition and similarities between the ML and biological systems.


## Comments
- I would suggest using the provided latex template
- Is there hope to improve PLMs (or machine learning models in general, e.g. having access to protein structure) to the extent where they would be able to discriminate all viral proteins? Or is there some intrinsic threshold due to the fact that these adapt and mimic human proteins?

---

### Official Review · Reviewer_QWLY · 2024-06-12
**-**

**Rating:** 7
**Confidence:** 5

**Review:**

-